

# Prognostic and therapeutic potential of gene profiles related to tertiary lymphoid structures in colorectal cancer

Jinglu Yu[1], Yabin Gong[2], Xiaowei Huang[3] and Yufang Bao[1]

[1] PuDong Traditional Chinese Medicine Hospital, Shanghai University of Traditional Chinese Medicine, Shanghai, Pudong New Area, China
[2] Department of Oncology, Yueyang Hospital of Integrated Traditional Chinese and Western Medicine, Shanghai University of Traditional Chinese Medicine, Shanghai, China
[3] Department of Oncology, Longhua Hospital, Shanghai University of Traditional Chinese Medicine, Shanghai, China

Corresponding author
Yufang Bao,
YufangBaosrly@126.com

## ABSTRACT

The role of tertiary lymphoid structures (TLS) in oncology is gaining interest, particularly in colorectal carcinoma, yet a thorough analysis remains elusive. This study pioneered a novel TLS quantification system for prognostic and therapeutic response prediction in colorectal carcinoma, alongside a comprehensive depiction of the TLS landscape. Utilizing single-cell sequencing, we established a TLS score within the Tumor Immune Microenvironment (TIME). Analysis of tertiary lymphoid structure-related genes (TLSRGs) in 1,184 patients with colon adenocarcinoma/rectum adenocarcinoma (COADREAD) from The Cancer Genome Atlas (TCGA) and Gene Expression Omnibus (GEO) databases led to the identification of two distinct molecular subtypes. Differentially expressed genes (DEGs) further segregated these patients into gene subtypes. A TLS score was formulated using gene set variation analysis (GSVA) and its efficacy in predicting immunotherapy outcomes was validated in two independent cohorts. High-scoring patients exhibited a 'hot' immune phenotype, correlating with enhanced immunotherapy efficacy. Key genes in our model, including *C5AR1*, *APOE*, *CYR1P1*, and *SPP1*, were implicated in COADREAD cell proliferation, invasion, and *PD-L1* expression. These insights offer a novel approach to colorectal carcinoma treatment, emphasizing TLS targeting as a potential anti-tumor strategy.

## INTRODUCTION

Colorectal cancer, a predominant gastrointestinal malignancy, ranks third in cancer-related mortality globally. The survival rates at 5 years hover around 65% (*Morgan et al., 2023*; *Siegel et al., 2023*). Surgical removal is the cornerstone for treating early-stage colorectal cancer, while advanced stages often benefit from an integrated approach of chemotherapy and targeted therapies (*Morris et al., 2023*). The advent of immune checkpoint inhibitors (ICIs) has marked a revolutionary shift in oncological therapeutics, offering significant clinical advantages for patients (*André et al., 2020*;

*Ludford et al., 2023*). Yet, the response to ICIs is limited to a small patient subset (*Gurjao et al., 2019*; *Marabelle et al., 2020*), underscoring the urgent need for biomarker research. This research is essential for refining patient selection and developing strategies to overcome immune resistance.

Historically, biomarker investigations have predominantly relied on RNA sequencing (RNA-Seq) from whole tumor tissues (*Thind et al., 2021*), providing a collective genetic snapshot of diverse cell populations. Unfortunately, biomarkers identified through this method have demonstrated restricted predictive capability. The emergence of single-cell RNA sequencing (scRNA-Seq) has revolutionized this landscape, facilitating gene expression analysis at an individual cell level and paving the way for the discovery of more effective biomarkers (*Kotsiliti, 2022*).

Tertiary lymphoid structures (TLS) are spontaneously formed ectopic lymphoid formations that arise in chronic inflammation sites, including tumor environments. These structures, which bear a structural similarity to secondary lymphoid organs, are chiefly comprised of an array of immune cells-B cells, T cells, dendritic cells, neutrophils, and macrophages (*Schumacher & Thommen, 2022*). Additionally, TLSs encompass high endothelial venules and lymphatic vessels, crucial for directing immune cell movement into TIME. Notably, the presence of TLSs has been associated with improved prognosis in a range of solid tumors, such as melanoma (*Cabrita et al., 2020*), renal cell carcinoma (*Meylan et al., 2022*), and colorectal cancer (*Overacre-Delgoffe et al., 2021*). Furthermore, emerging studies highlight a significant correlation between TLSs and the efficacy of immunotherapy (*Helmink et al., 2020*). The existence of TLSs has been identified as a predictive biomarker for the response to ICI therapy in advanced-stage cancers, including bladder cancer (*Groeneveld et al., 2021*) and head and neck squamous cell carcinoma (*Ruffin et al., 2021*). Consequently, the induction of TLSs is increasingly being explored as a novel therapeutic approach in oncology (*Vanhersecke et al., 2021*).

Additionally, numerous investigations have demonstrated that prevalent anti-cancer therapies are capable of prompting intratumoral TLS formation in murine models. *Chelvanambi et al. (2021)* utilized the STING agonist ADU S-100 in treating B16.F10 melanomas, led to STING activation within the TIME, correlating with the reduction of melanoma progression and concurrent development of TLSs. *Lee et al. (2022)* employed stromal vascular fraction spheroid-based immunotherapy, which resulted in the formation TLS-like structures, thereby enhancing antigen-specific immune responses and anti-tumor immunity in mice. While immunohistochemistry remains the conventional method for TLS detection (*Buisseret et al., 2017*), there is an escalating need for more definitive approaches to assess TLS levels through transcriptomic analysis of tumor biopsies or excised samples. A gene signature encompassing *CCL2*, *CCL4*, *CCL5*, *CCL8*, *CXCL9*, *CXCL10*, *CXCL11*, *CXCL13*, *CCL18*, *CCL19*, and *CCL21* has been formulated and employed to classify tumors as either TLS+ or TLS- (*Calderaro et al., 2019*). Tumors categorized as TLS+ have shown a reduced risk of early recurrence in hepatocellular carcinoma (*Calderaro et al., 2019*). This has led to our initiative to establish a refined TLS signature for colon adenocarcinoma/rectum adenocarcinoma (COADREAD), aiming to

analyze TLS neogenesis through a multi-omics lens and thereby enhance the scope of future clinical research.

Nevertheless, the intricacies of how TLSs interact with COADREAD, as well as the interplay among immune cells within TLSs, especially on the single cell level, remain enigmatic. Moreover, the contribution of TLSs in modulating responses to immune therapy is yet to be fully elucidated.

In this research endeavor, our objective was to amalgamate single-cell and bulk RNA sequencing data to elucidate the role of TLSs, particularly their potential as biomarkers for predicting ICI outcomes in patients with COADREAD. Additionally, our study sought to unravel the molecular and immune characteristics of TLSs, assessing their impact on the prognosis for COADREAD patients. To this end, we have devised TLS Score, which could serve as an invaluable tool in the realm of personalized precision medicine, aiding clinicians in tailoring treatment strategies.

## MATERIALS AND METHODS

### Data sources and analysis workflow

The study's analytical process was depicted in Fig. 1. We obtained the CRC_EMTAB8107 scRNA-seq dataset for COADREAD from the Gene Expression Omnibus (GEO) database (https://www.ncbi.nlm.nih.gov/geo/) (*Barrett et al., 2013*), which includes seven instances of colorectal carcinoma. Furthermore, The Cancer Genoma Atlas (TCGA) database (https://portal.gdc.cancer.gov) was used to obtain the bulk RNA-seq data and clinical profiles of 380 COADREAD patients (*Blum, Wang & Zenklusen, 2018*). We also sourced datasets GSE17538 ($n = 238$) and GSE39582 ($n = 566$) from the GEO repository.

### Single-cell dataset processing and cluster identification

For the scRNA-seq dataset, we employed Seurat (v4.1.1) in the R programming environment (*Mangiola, Doyle & Papenfuss, 2021*). Initial steps included quality assessment using the "Seurat" package, leading to the exclusion of cells not meeting specific criteria: (1) nFeature_RNA range of 200 to 7,500; (2) nCount_RNA range of 200 to 35,000; (3) a mitochondrial gene content of ≥10%. This filtration yielded a dataset comprising 66,050 cells and 21,753 genes. Following actions involved batch normalization using the 'harmony' package and data scaling with the 'ScaleData' function. The top 28 principal components were extracted by performing principal component analysis (PCA) on the 2,000 genes with the highest variability. UMAP was used for unsupervised clustering to visualize clusters on a two-dimensional plane (*Becht et al., 2018*). The function 'FindAllMarkers' was used to evaluate gene expression variations among clusters, considering the criteria of |log2 (fold change)| > 1, an adjusted $p$-value < 0.05, and a resolution of 0.5. Cell subpopulations were annotated employing the "SingleR" package, CellMarker database (*Hu et al., 2023*), and PanglaoDB database (*Franzén, Gan & Björkegren, 2019*).

Following this, the "FindAllMarkers" function in the R environment was employed to identify genes differentially expressed across the various cell types. We used the GSVA package (version 1.40.1) to calculate GSVA scores for 50 hallmark gene sets obtained from

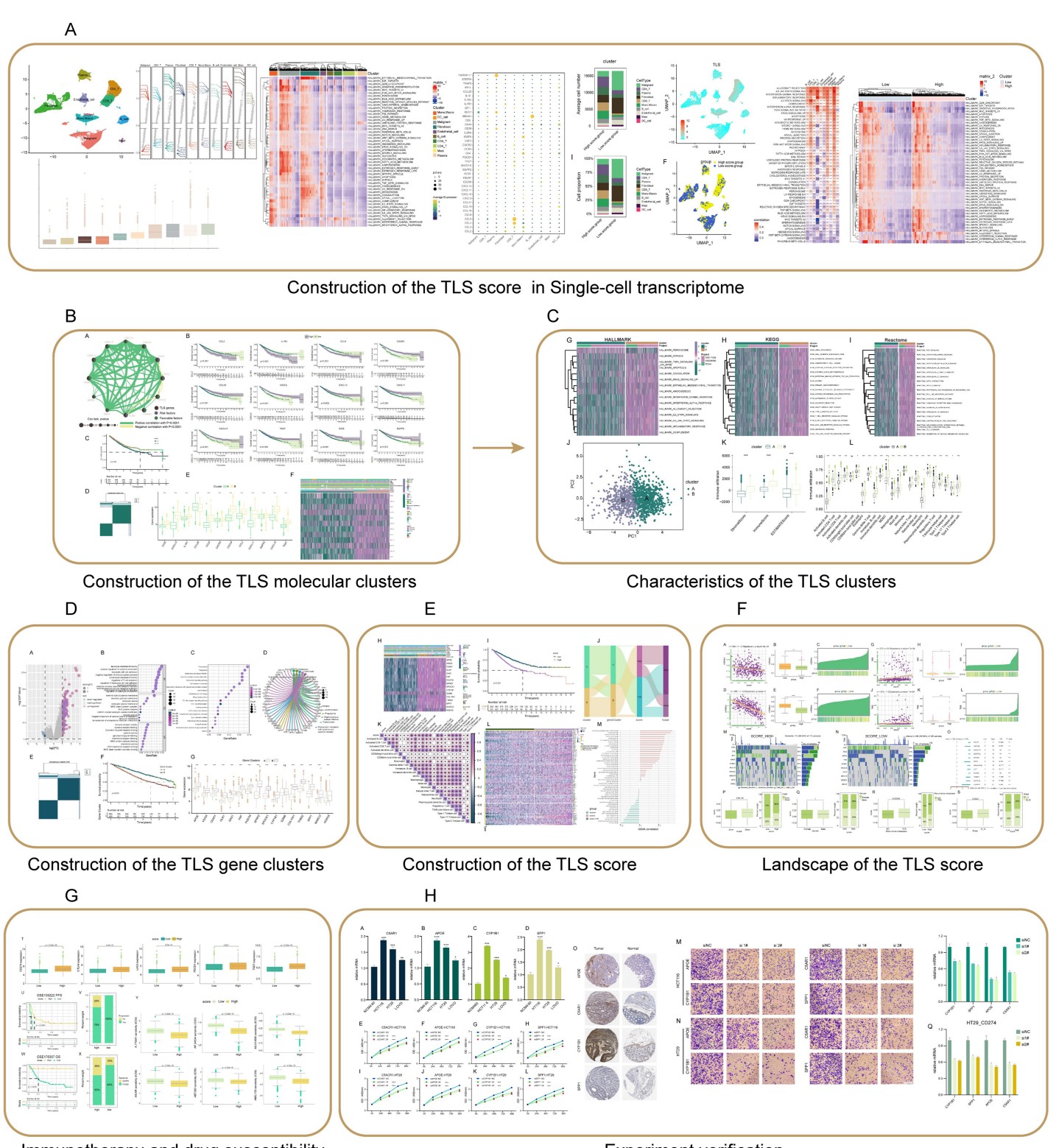

Figure 1 Flowchart of this study. (A) The CRC_EMTAB8107 scRNA-seq dataset was downloaded from the GEO Database (https://www.ncbi.nlm.nih.gov/geo/), and the hallmark pathways were retrieved from the MsigDB database (https://www.gsea-msigdb.org/gsea/msigdb/index.jsp). (B–E) The GSE17538 and GSE39582 datasets were downloaded from the GEO database, and the COADREAD dataset was retrieved from the TCGA database (https://portal.gdc.cancer.gov). Hallmark, KEGG, and Reactome pathways were all sourced from the MsigDB database. (F) The datasets for
**Figure 1 (continued)**
analyzing MSI, TMB, CSCs, and somatic mutations were sourced from the TCGA_COADREAD database. (G) The immunotherapy datasets GSE135222 and GSE176307 were obtained from the GEO database. Drug sensitivity analyses were performed using the GSE17538, GSE39582 datasets from the GEO database, the COADREAD dataset from TCGA, and data from the CancerRxGene database (https://www.cancerrxgene.org/). (H) The immunohistochemistry results were retrieved from the HPA database (https://www.proteinatlas.org).

the Molecular Signatures Database (MSigDB) (https://www.gsea-msigdb.org/gsea/msigdb) (*Liberzon et al., 2015*). Concurrently, we obtained a set of 39 TLSRGs as outlined in PMID: 31092904 (*Sautès-Fridman et al., 2019*), and applied the GSVA package to calculate their scores. Based on the median score, cell clusters were segregated into groups with high and low TLS scores.

Furthermore, to discern the relationship between TLS and hallmark scores, and to analyze the clustering of TLS scores with hallmark pathways, we used a heat map representation. The creation of heat maps and correlation heat maps was facilitated by the "pheatmap" and "corrplot" packages. All threshold values for these analyses were set as per the default specifications detailed in the package vignettes.

## Elaborate analysis of bulk RNA-sequencing data
### Formulation of the prognostic TLSRGs signature
We initiated our analysis by amalgamating the bulk RNA-seq datasets, rectifying batch effects using the "limma" and "sva" packages. Subsequently, a log2 transformation was applied to standardize the data. The relationship between TLSRGs expression levels and patient overall survival (OS) was scrutinized using univariate Cox regression analysis, utilizing the "survival" package in R (version 3.2–13) (https://CRAN.Rproject.org/package=survival).

### Consensus clustering for prognostic TLSRGs
We used the R package 'ConsensusClusterPlus' to perform unsupervised clustering analysis on COADREAD patients, stratifying them based on the expression of prognostic-TLSRGs. By employing the K-means algorithm, we identified the optimal number of clusters ranging from 2 to 10. This process was reiterated 1,000 times to affirm the stability and reliability of our results.

### Clinicopathological traits
Investigating the clinical implications, we analyzed the correlation between the molecular subtypes and clinicopathological features, subsequently visualized in a heatmap. The clinicopathological features that were examined included the patient's age, sex, tumor grade, and stage of tumor node metastasis (TNM). The "survminer" packages in R facilitated our survival analysis.

### Pathway enrichment analysis
To elucidate the TIME characteristics in various molecular subtypes, GSVA was employed utilizing gene sets from hallmark, KEGG, and Reactome obtained from the MSigDB database.
### Immune infiltration

The stromal score, immune score, and ESTIMATE score in tumor tissues were quantified using the "estimate" R package (*Scire et al., 2023*) by employing the ESTIMATE algorithm from the ESTIMATE project (https://sourceforge.net/projects/estimateproject/). Furthermore, we utilized ssGSEA to delineate the immune infiltration pattern in COADREAD by employing marker genes for 23 immune cell types (*Hänzelmann, Castelo & Guinney, 2013*).

### Analysis of differentially expressed genes

The "limma" package was utilized to identify differentially expressed genes (DEGs) among clusters, specifically targeting those with an adjusted *p*-value < 0.05 and |log2 (Fold Change)| >1. The "clusterProfiler" R package was used to perform Gene Ontology (GO) and Kyoto Encyclopedia of Genes and Genomes (KEGG) analyses. To discern DEGs with prognostic significance, univariate Cox regression analysis was executed, with results presented in a forest plot. Additionally, the GSCA datasets (https://guolab.wchscu.cn/GSCA/#/) (*Liu et al., 2023*) were used to perform mutation, SNV, CNV, and Methylation analyses on these prognostic genes.

### Refinement of the TLS score system

We employed the "ConsensusClusterPlus" package in R for unsupervised consensus clustering to categorize COADREAD patients into distinct molecular subtypes. The classification was determined by analyzing the expression patterns of previously identified prognostic TLS-DEGs. The clustering process involved the K-means algorithm, assessing optimal cluster numbers from k = 2–10. This procedure was iteratively performed 1,000 times for robust stability of results. Kaplan-Meier plots elucidated significant prognostic variations between the clusters. Additionally, a heatmap delineated the interplay between clinical attributes, gene expression profiles, and clustering outcomes. The TLS score, computed *via* the GSVA package using 16 pivotal prognostic TLS-DEGs, facilitated subsequent survival analysis. This analysis discerned the prognostic predictive capability of the TLS score in COADREAD cases. A Sankey diagram illustrated the intricate associations among signature genes, scoring, and prognostic categorization.

### Comprehensive immune landscape analysis utilizing the TLS score

To further understand the TLS score's implications, we created a correlation heatmap showcasing its association with immune cell infiltration levels. Moreover, the interrelations between the TLS score and various cytokines, chemokines, and their receptors were visualized through additional heatmaps. The GSVA method was pivotal in correlating the TLS score with 50 hallmark pathways, with the findings depicted using R's "ggplot2" package. We also gathered and analyzed mRNA expression data of immune checkpoints and gene mutation information from the TCGA-COADREAD cohorts. This led to the calculation of mean normalized expression levels of immune checkpoints per sample, subsequently normalized to convey their relative expression magnitudes.

### Analysis of MSI, TMB, CSCs, and somatic mutations

We conducted an in-depth analysis of MSI, CSC markers, and TMB across varying TLS score groups. For this, somatic mutation data specific to COADREAD patients were extracted from the TCGA database. To discern somatic mutations between different TLS score groups, we utilized the "maftools" package to construct detailed waterfall plots. A comparative analysis of the somatic mutation variances across the TLS score groups was then conducted using forest plots.

### Evaluation of immunotherapy response and chemotherapy sensitivity

The predictive relevance of the TLS score for immunotherapy response was investigated using transcriptomic data from two distinct cohorts. The dataset GSE135222 includes information from patients with advanced non-small cell lung carcinoma who received treatments targeting anti-PD-1/PD-L1. Meanwhile, GSE176307 includes transcriptomic information from patients with metastatic urothelial cancer. The prognostic significance of the TLS score in COADREAD was evaluated using Kaplan-Meier analysis in the R package. Moreover, by utilizing the CancerRxGene database (https://www.cancerrxgene.org/) (*Yang et al., 2013*), we calculated the IC50 values for different target-therapeutic medications in relation to the TLS score. This estimation was performed using the "pRRophetic" package in R, offering insights into the chemotherapy sensitivity linked to the TLS score.

### RNA extraction and quantitative real-time PCR

Total RNA from colorectal cells was isolated employing the Trizol method (Invitrogen, Waltham, MA, USA) as per the manufacturer's instructions. Subsequent reverse transcription to synthesize cDNA was conducted using Prime Script RT reagent Kit (RR047A, Takara, Shiga, Japan). Gene expression was measured using TB Green Premix Ex Taq (RR420A, Takara, Shiga, Japan), and GAPDH was used as the reference for normalization. Details of the primers used are provided in Table S1.

### Cultivation of cell lines and reagent preparation

The Chinese Academy of Sciences Cell Bank (Shanghai, China) provided several colorectal cancer cell lines, namely NCM460, HCT116, HT29, and LOVO. Each cell line underwent STR profiling for authentication and was confirmed to be free of mycoplasma contamination (https://www.cellbank.org.cn/). NCM460 and HCT116 were cultured in RPMI 1640 medium (Gibco, Waltham, MA, USA) with the addition of 10% FBS, whereas HT29 and LOVO were maintained in DMEM (Gibco, Waltham, MA, USA) with a 10% FBS enrichment. Cultures were incubated at 37 °C in a humidified atmosphere with 5% $CO_2$.

### Targeted knockdown of TLS-related genes

To conduct gene silencing experiments, HCT116 and HT29 cells were plated at a density of $3 \times 10^5$ cells per 60 mm dish. Following 24 h of incubation, the medium was replaced with fresh growth medium. Next, the cells were transfected with siRNAs that targeted C5AR1, APOE, CYP1B1, and SPP1, or a control siRNA obtained from Genepharma in Shanghai,

China. This transfection was performed using Lipofectamine RNAiMax (13778075; Life Technologies, Carlsbad, CA, USA). After transfection, the cells were cultured in RPMI 1640 or DMEM, with the addition of 10% FBS, for at least 24 h. The siRNA sequences are listed in Table S1.

### Cell proliferation assessment

Cells were seeded in 96-well plates at a concentration of 3,000 cells per well. The growth of cells was observed for a period of 24 h using a Cell Counting Kit-8 (CCK-8) from Dojindo, following the manufacturer's provided instructions. A microplate reader (Thermo Fisher, Waltham, MA, USA) was used to acquire absorbance measurements at 450 nm. GraphPad Prism version 9.5.0 was utilized for conducting data analysis.

### Transwell migration assay

To perform migration experiments, a total of 100,000 cells were placed in the top compartment of a Transwell device on a Matrigel layer (Corning, Corning, NY, USA), utilizing either RPMI 1640 or DMEM excluding FBS. The bottom section contained 600 µL of the corresponding solution enriched with 20% FBS. After a 24-h incubation period, cells were fixed and stained with crystal violet. Cells remaining in the upper chamber were carefully removed, and the migrated cells were imaged for further analysis.

### Statistical methodology

The statistical analysis was conducted using R software (version 4.1.2). Correlation assessments were conducted using Pearson or Spearman correlation coefficients. The Wilcoxon test was applied for comparisons between two groups. The overall survival (OS) across various subgroups was analyzed using Kaplan-Meier survival curves and log-rank tests. Univariate Cox regression analyses were utilized to estimate prognostic values. The data from the qRT-PCR analysis were examined utilizing the t-test of Student. A $p$-value less than 0.05 was deemed to be statistically significant, with an asterisk (*) representing $p$-values less than 0.05, two asterisks (**) representing $p$-values less than 0.01, and three asterisks (***) representing $p$-values less than 0.001.

### Ethical consideration

The authors conducted this study without the participation of any humans or animals.

## RESULTS

Flowchart of this study was shown in Fig. 1.

### Analysis of single cell sequencing data
### Dimensionality reduction

In our initial step, the single-cell sequencing data from the CRC_EMTAB8107 dataset, were subjected to an integration analysis to consolidate information across different samples, Fig. S1 exhibited the minimal batch effects and 19 distinct clusters categorized by the UMAP algorithm. Further, we examined the expression of specific surface marker genes across these clusters. This analysis enabled the identification of 10 unique cell types, encompassing malignant cells, CD4+ T cells, plasma cells, fibroblasts, CD8+ T cells,

monocytes-macrophages, B cells, endothelial cells, mast cells, and dendritic cells, as depicted in Fig. 2A.

### Landscape of transcriptome

In the realm of gene expression, Fig. 2B highlighted the top five genes making the most significant contributions. Specifically, in malignant cells, the genes exhibiting the highest expression levels included *C19orf33*, *KRT19*, *TSPAN8*, *AGR2*, and *LGALS4*. In stark contrast, genes such as *SRGN*, *RGS1*, *TSC22D3*, *CXCR4*, and *CD52* were among the least expressed. Within CD8+ T cells, genes like *CCL5*, *NKG7*, *GZMA*, *CCL4*, and *GNLY* were highly expressed, while others including *CALD1*, *CST3*, *SPARCL1*, *GSN*, and *IFITM3* showed minimal expression. For monocytes-macrophages, the top expressed genes were *AIF1*, *TYROBP*, *C1QA*, *FCER1G*, and *C1QB*, whereas genes like *TRAC*, *CALD1*, *IL32*, *CD69*, and *FKBP11* were the least expressed.

### Pathway enrichment analysis

The GSVA brought to light hallmark pathways that were predominantly enriched in each cell type (see Fig. 2C). In malignant cells, pathways such as mTORC1 signaling, glycolysis, xenobiotic metabolism, fatty acid metabolism, were notably prevalent. Fibroblasts showed a significant enrichment of the epithelial mesenchymal transition pathway. The monocytes-macrophages were enriched in pathways such as complement, IL2-STAT5 signaling, inflammatory response, and interferon response pathways.

### Cell-type specific expression of TLS insights

Employing the "FindAllMarkers" function, a total of 39 TLSRGs were analyzed. This detailed examination culminated in deciphering the expression patterns of TLSRGs across various cell types (Fig. 2D). Focusing on cell-specific expression, *MS4A1* emerged as a gene predominantly expressed in B cells, while its expression remained minimal in other cell types. In the realm of CD8+ T cells, genes like *CCL4* and *CCL5* showcased high expression levels, contrasting with their subdued expression in other cells. Additionally, *TNFRSF17* was notably expressed in plasma cells, yet exhibited low expression in other cell types.

### GSVA-based TLS score calculation

Utilizing the UMAP cell annotation map, we calculated the TLS score for each cell through the GSVA method, as illustrated in Fig. 2E. Subsequently, cells were categorized into clusters with low and high TLS scores based on their median expression levels, depicted in Fig. 2F. Predominantly, malignant cells and fibroblasts were found in the low TLS score cluster, whereas T cells and monocyte-macrophages were more prevalent in the high TLS score cluster.

### Cell cluster distribution and TLS score visualization

This distribution was further detailed through box plots representing TLS scores across different cell clusters (Fig. 2G), where CD8+ T cells received the highest scores, and mast cells the lowest. The distribution patterns of cell types within these clusters were effectively represented in a composition chart based on cell count and proportion (Figs. 2H and 2I), highlighting the tendency of malignant cells and fibroblasts to fall into the low scoring

![PeerJ]

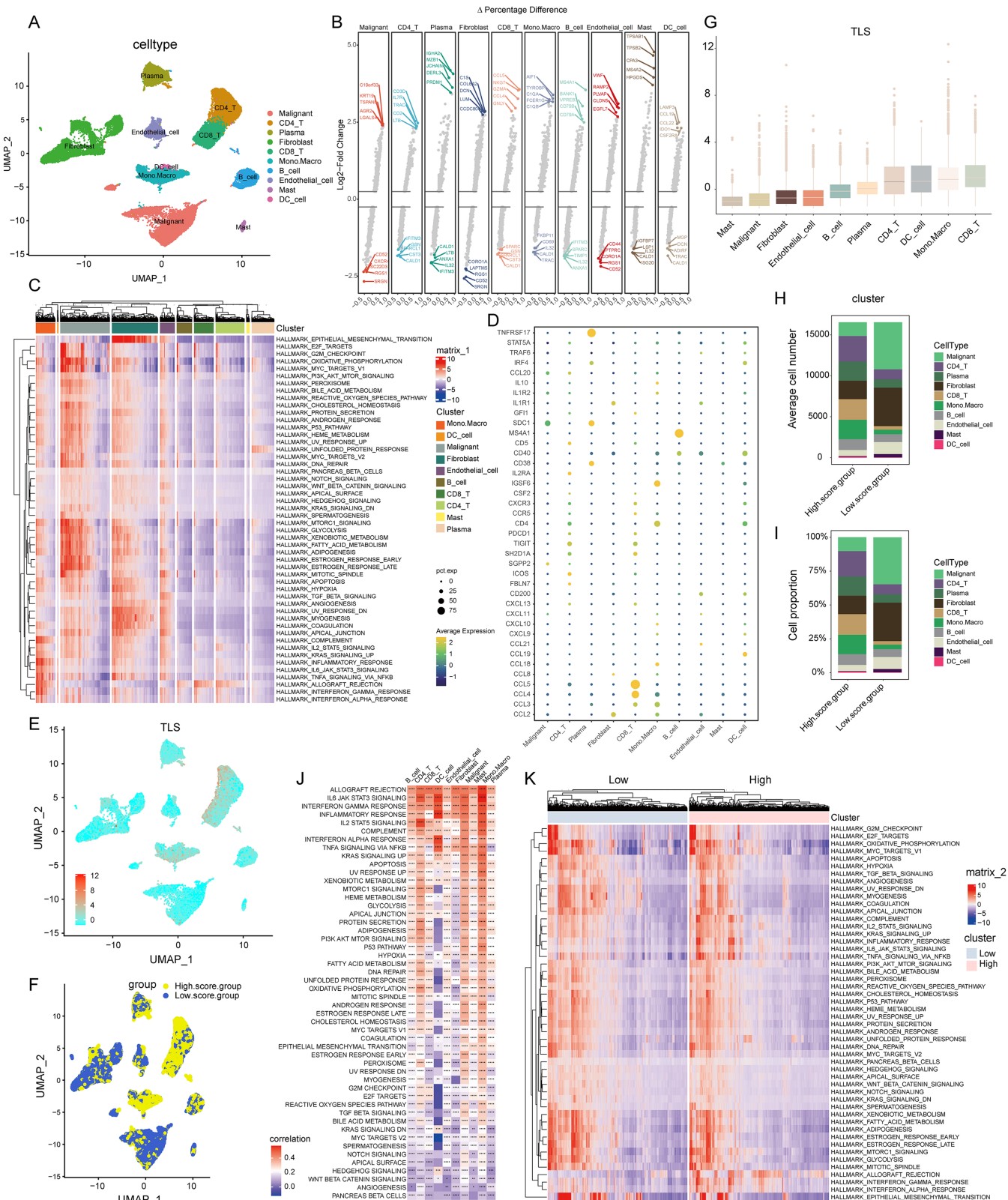

**Figure 2** **Construction of TLS score revealing high cellular heterogeneity in COADREAD based on single cell RNA sequencing (scRNA-seq) data.** (A) Reduced dimensionality and cluster analysis. The COADREAD dataset cells might be classified into 19 clusters by UMAP, which included B cells, dendritic cells, endothelial cells, fibroblasts, malignant cells, mono-macrophages, mast cells, plasma cells, CD8+T cells and

**Figure 2 (continued)**
CD4+T cells. (B) The volcano plot displaying differential expression genes across various cell types, with the top five shown. (C) The heatmap displayed the score of significantly enriched hallmark pathways across each tumor infiltrating cell type. The bar represented the row-scaled pathway enrichment level. (D) The dot plot displayed the expression levels of 39 TLSRGs across various cell types. (E) Identification and landscape of TLS score. The TLS score of each cell type was calculated by GSVA. The expression of TLS gene cluster in each cell type was visualized by UMAP plot. (F) Grouping by TLS score. The cell clusters were divided into high- and low score groups and displayed by UMAP plot. (G) Comparison of TLS score among cell clusters. Box plots showed the TLS score of every cell cluster and were sorted by ascending order. (H–I) Quantization of cell number and proportion in TLS score clusters. Cell number and proportion of each cell cluster was compared between high- and low- TLS groups by composition chart. (J) Correlation of TLS score with hallmark pathways. The heatmap showed the correlation of TLS score with hallmark pathway scores across all cell clusters. (K) Enrichment of hallmark pathways. The comparison of hallmark pathway score between high- and low-score group was showed by heatmap.                               

group, in contrast to monocyte-macrophages and CD8+ T cells which were more commonly found in the high scoring group.

### TLS score correlation with hallmark pathways

Further analysis revealed a significant correlation between the TLS score and specific hallmark pathway scores (Fig. 2J). In malignant cells, the TLS score showed a positive correlation with pathways such as apoptosis and KRAS signaling up. In monocyte-macrophages, a positive correlation was observed with the Inflammatory response, interferon gamma response, IL2-STAT5 signaling pathways. For dendritic cells, the TLS score positively correlated with pathways including Inflammatory response, Interferon gamma response, Interferon alpha response. We identified hallmark pathways that were differentially expressed between high- and low-TLS score clusters, as shown in Fig. 2K. These enriched pathways in high-score group included inflammatory response, interferon gamma response, interferon alpha response, markedly distinct compared to the low-TLS score group.

## Comprehensive analysis of TLSRGs in COADREAD

### TLS molecular subtypes: identification and analysis

Our exploration focused on discerning the TLS patterns in COADREAD tumorigenesis. To this end, we examined 1,184 COADREAD patient samples sourced from the TCGA and GEO databases (including GSE39582 and GSE17538). Survival analysis pinpointed 12 TLSRGs (*CCL2, CCL20, CXCR3, IL1R1, TIGIT, SGPP2, ICOS, CCL8, CXCL9, CXCL11, CXCL10, CXCL13*) significantly correlated with the overall survival (OS) in COADREAD patients (Fig. 3B, $p < 0.05$). We then constructed a TLS network to delineate the interrelationships among TLSRGs and their prognostic values (Fig. 3A). Univariate Cox regression analysis on these genes revealed positive associations of *CCL20, CXCR3, TIGIT, SGPP2, ICOS, CXCL9, CXCL11, CXCL10*, and *CXCL13* with patient survival, while *CCL2, CCL8*, and *IL1R1* showed negative associations.

Based on the 12 prognostic TLSRGs, we employed a consensus clustering algorithm to divide COADREAD patients into two distinct molecular subtypes, namely Cluster A and Cluster B, with the analysis suggesting k = 2 as the optimal cluster number (Fig. 3D). Patients in subtype B exhibited significantly higher survival rates compared to those in subtype A, as demonstrated by the log-rank test ($p = 0.001$; Fig. 3C). A notable divergence

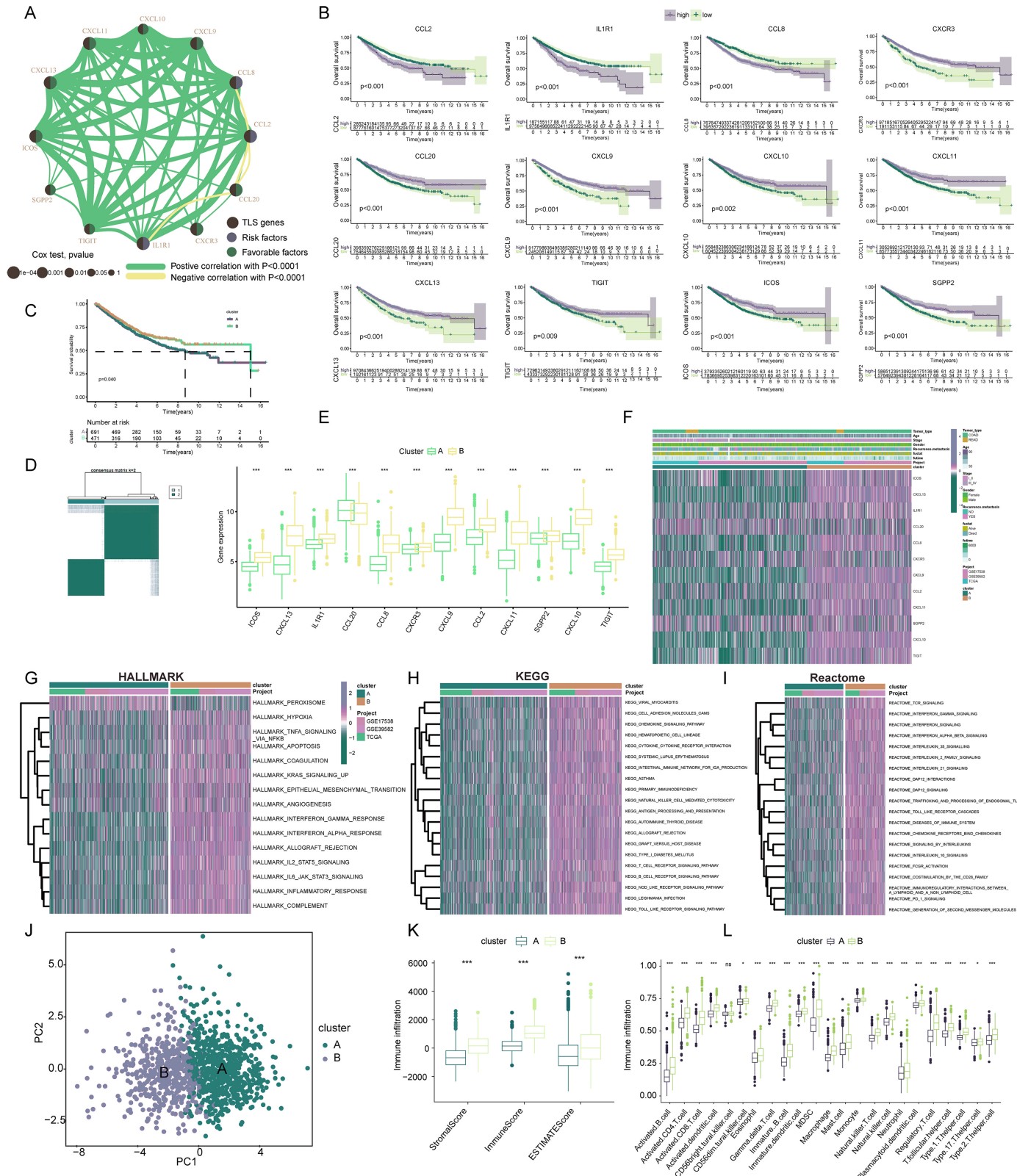

**Figure 3** **Construction and biological analysis of TLS molecular subtypes in colorectal carcinoma.** (A) UniCox regression analysis and mutual correlations among TLSRGs in 1,184 colorectal cancer samples. Spearman correlation analyses were used. The line between two TLSRGs indicated
**Figure 3 (continued)**
their interaction, and the stronger the correlation, the thicker the line. green line represented positive correlation and yellow line represented negative correlation. (B) Kaplan–Meier plot and log-rank tests were conducted for survival analyses of those TLSRGs (CCL2, CCL20, CXCR3, IL1R1, TIGIT, SGPP2, ICOS, CCL8, CXCL9, CXCL11, CXCL10, CXCL13). (C) Survival analyses for two different clusters. (D) Consensus clustering matrix for k = 2. (E) The abundance of TLSRGs in two clusters. (F) The heat-map showed the associations of clinicopathologic characteristics with molecular subtypes. Purple color indicated up-regulation and green color indicated down-regulation. GSVA enrichment analyses of Hallmark (G), KEGG (H), and Reactome (I) pathways in subtype A and B. Purple color indicated more enriched pathways and green color indicated less enriched pathways. (J) PCA presented a great difference in transcriptomes between different TLS molecular subtypes. (K) Differences in the stromal, immune and ESTIMATE score between different TLS clusters. (L) The abundance of tumor infiltrating immune cells in cluster A and B. * indicates a significant difference between the two groups at $p < 0.05$. *** indicates a highly significant difference between the two groups at $p < 0.001$.

in the expression patterns of TLSRGs was observed between clusters A and B (Fig. 3E). Particularly, a subset of TLSRGs, including *CXCL9*, *CXCL10*, *CXCL11*, *CXCL13*, *ICOS*, *TIGIT*, among others, were found to be predominantly upregulated in cluster B ($p < 0.05$). Further, we profiled the associations of TLSRG expression with various clinical characteristics such as age, disease stage, gender, recurrence, metastasis, fustat, and futime across the different subtypes (Fig. 3F). This profiling provided insights into the clinical relevance of TLSRG expression variations in distinct patient subgroups. To enhance the distinct biological traits of the subtypes, our study incorporated a comprehensive GSVA enrichment analysis using Hallmark, KEGG, and Reactome pathway databases. Collectively, these comprehensive pathway analyses consistently underscored the prominence of interferon gamma response, interferon alpha response, chemokine signaling pathway, and cytokine-cytokine receptor interaction pathways in subtype B. These pathways were critically linked to immune activation (*Yenyuwadee et al., 2022*), further delineating the distinct immunological landscape of subtype B (Figs. 3G–3I).

The PCA analysis reinforced the significant transcriptional distinctions between subtype A and B (Fig. 3J). To delve deeper into the role of TLSRGs in shaping the TIME, we quantified human immune cell subsets in each COADREAD sample. As depicted in Fig. 3K, cluster B was characterized by elevated stromal, immune, and ESTIMATE scores. A more granular examination through ssGSEA revealed marked disparities in immune cell infiltration between subtypes A and B (Fig. 3L). Specifically, subtype B demonstrated notably higher infiltration levels of activated B cells, activated CD4+ T cells, activated CD8 + T cells, activated dendritic cells, macrophages, natural killer T cells, natural killer cells, T follicular helper cells, Type 1 T-helper cells, and Type 2 T-helper cells compared to subtype A.

From these observations, it can be inferred that subtype B was significantly enriched in immune activation pathways, which appears to be intricately linked with the dynamics of TIME. In the quest to decode the biological intricacies of TLS subtypes, we scrutinized 189 DEGs between subtypes A and B. These were visually represented through a volcano plot (Figs. 4A, |logFC| > 1, $p < 0.05$). To identify pertinent biological pathways, we engaged in comprehensive GO and KEGG enrichment analyses. The GO and KEGG enrichment analysis pinpointed the DEGs' significant involvement in pathways such as cytokine-cytokine receptor interaction, upregulation of cytokine production,

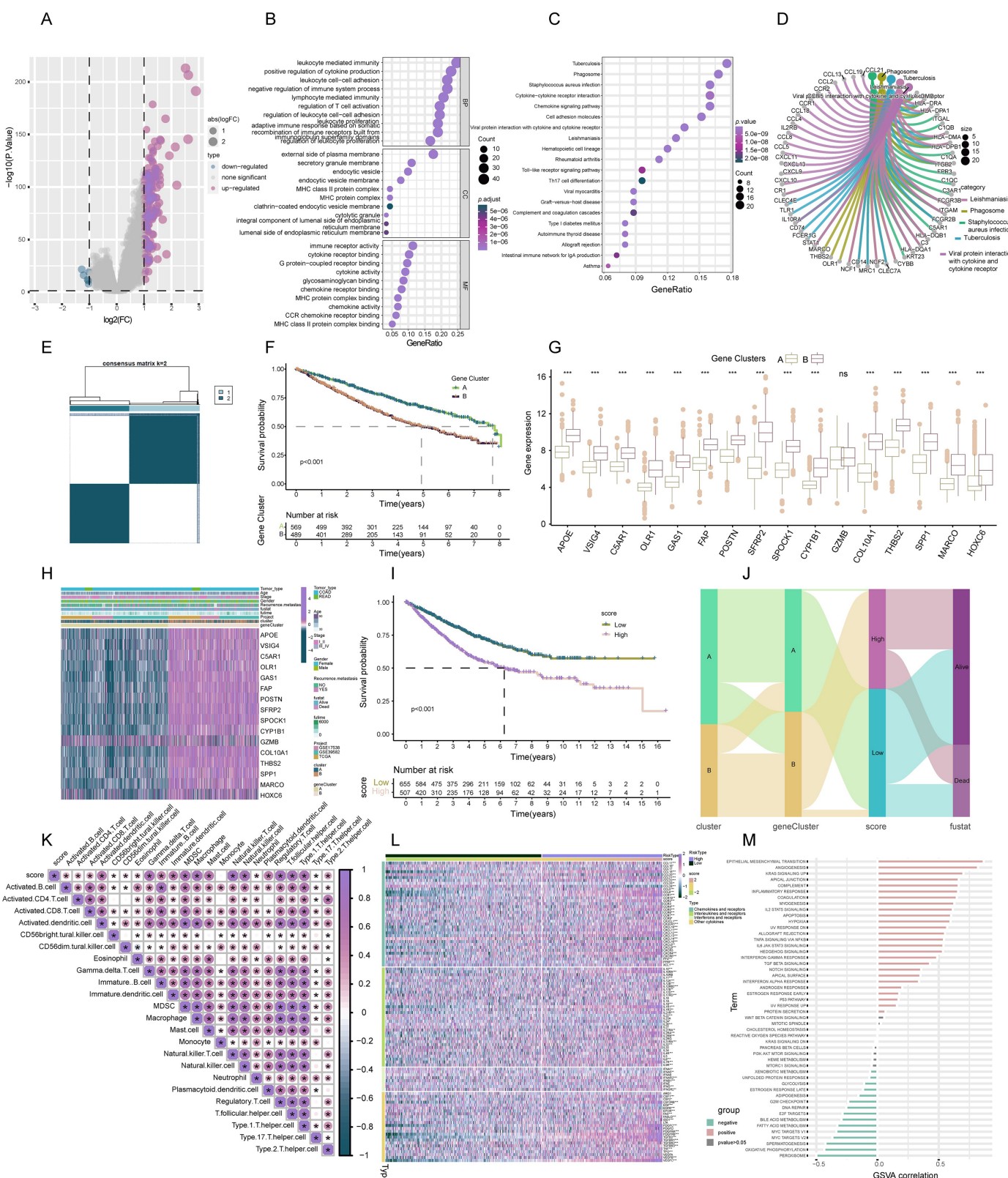

**Figure 4 Construction of TLS gene subtypes and TLS score, following integrative analysis.** (A) 189 TLS-DEGs shown in the volcano plot. (B) Functional annotation for TLS-DEGs using GO enrichment analysis. The size of the plots represented the number of genes enriched. (B) The

**Figure 4 (continued)**
pathways were grouped by biological process (BP), cellular component (CC), and molecular function (MF). (C) Functional annotation for TLS-DEGs using KEGG enrichment analysis. The size of the plots represented the number of genes enriched. (D) The chord graph showed the 5 vital pathways and corresponding genes of KEGG analysis. (E) Identification of two gene subtypes (k = 2) and their correlation area through consensus clustering analysis of the 16 genes. (F) Survival analysis of the two gene subtypes. (G) Expression of the 16 genes in different subtypes. (H) The heat-map showed the associations of clinicopathologic characteristics with gene subtypes. (I) TLS scores were calculated based on the 16 TLS-DEGs by GSVA, and survival analyses were performed in high- and low-score group. (J) Sankey diagram displayed the relationship of molecular subtypes, gene subtypes, TLS scores and survival outcomes. (K) The TLS score was positively associated with all over tumor infiltrating immune cells. (L) GSVA analysis for cytokines, chemokines, and their receptors in high- and low-TLS score clusters. (M) Correlation of TLS score with 50 hallmark pathway scores by GSVA. *** indicates a significant difference between the analyzed variables at $p < 0.001$.

lymphocyte-mediated immunity, and adaptive immune responses founded on somatic leukocyte proliferation (Figs. 4B, 4C). A cnetplot was constructed to visualize the interplay within these pathways, highlighting the top five pathways (Fig. 4D).

### Identification of TLS gene subtypes

In the continuum of our research, a univariate Cox regression analysis was conducted to evaluate the prognostic significance of the 189 identified DEGs. This rigorous analysis led to the isolation of 16 key signature genes closely associated with overall survival (OS) in COADREAD patients ($p < 0.001$). These genes, namely *HOXC6, SFRP2, CYP1B1, GAS1, SPOCK1, SPP1, POSTN, MARCO, VSIG4, THBS2, OLR1, COL10A1, FAP, GZMB, C5AR1,* and *APOE*, were depicted in a forest plot for clarity (Fig. S2A).

Subsequently, COADREAD patients were stratified into two distinct groups based on the expression profiles of these 16 prognostic genes, utilizing a consensus clustering algorithm. The partitioning suggested that k = 2 was an optimal division, forming two gene clusters, namely A and B (Fig. 4E). Survival analysis revealed a noteworthy finding: patients in subtype B exhibited a significantly enhanced survival probability compared to those in subtype A (log-rank test, $p < 0.001$; Fig. 4F). The box-map analysis highlighted that most of the signature genes, including *APOE, VSIG4, C5AR1, CYP1B1,* and *SPP1*, were predominantly 11 upregulated in cluster B (Fig. 4G, $p < 0.05$). Moreover, we extended our analysis to explore the correlation of these gene expressions with various clinical features such as age, stage, gender, recurrence, metastasis, fustat, and futime, stratifying them across the molecular subtypes (Fig. 4H).

### Construction of TLS score

In advancing our study, a TLS score was formulated based on GSVA of the 16 identified signature genes. This scoring system facilitated the stratification of patients into two distinct genomic subtypes: those with low and high TLS scores. Survival analysis revealed a critical insight; patients categorized in the low-score group exhibited a markedly higher survival probability compared to their high-score counterparts (Fig. 4I, $p < 0.001$). A Sankey diagram was employed to eloquently depict the patient distribution across the two TLS molecular subtypes, gene subtypes, and score groups, alongside their respective prognostic implications (Fig. 4J).

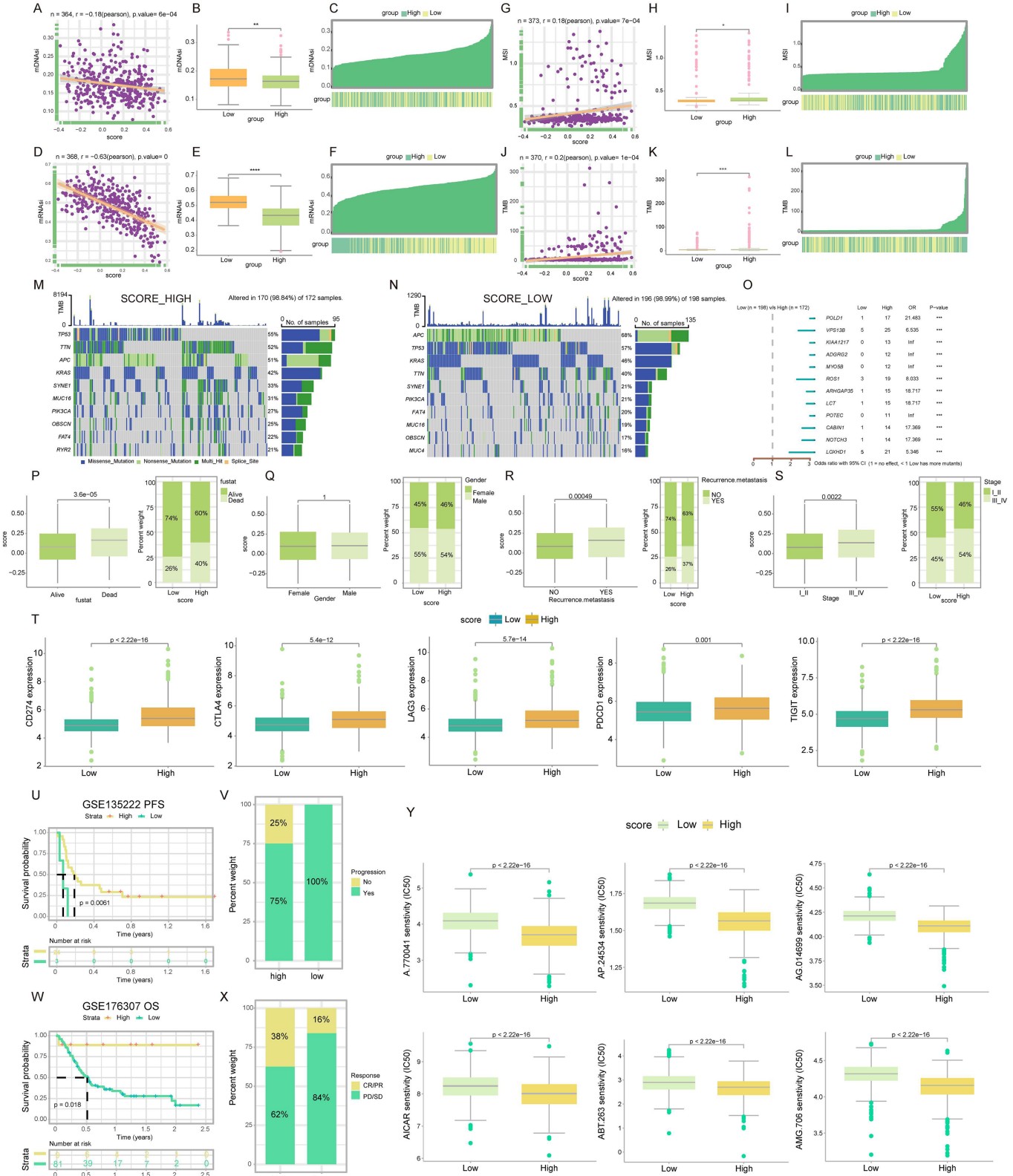

**Figure 5** **Mutations landscape, clinical significance, immunotherapy response and drug susceptibility of TLS score.** Correlation (A), expression (B) and the distribution (C) of TLS score with mDNAsi index. Correlation (D), expression (E) and the distribution (F) of TLS score with mRNAsi

**Figure 5 (continued)**
index. Correlation (G), expression (H) and the distribution (I) of TLS score with MSI index. Correlation (J), expression (K) and the distribution (L) of TLS score with TMB index. The waterfall plot of somatic mutation characteristics in high-and low-TLS score groups. High score group contained 172 COADREAD samples (M) and low score group contained 198 COADREAD samples (N). (O) Mutational variation in high-and low-score clusters, Forest plot shows the HRs of two score clusters. Differences between TLS score clusters in fustat (P), gender (Q), recurrence/metastasis (R), stage (S) and the expression of immune checkpoints (T). Difference of survival analyses (U), response to anti-PD-1/PD-L1 therapy (V) between low- and high-TLS clusters in non-small cell lung carcinoma cohort (GSE135222). Differences of survival analyses (W), response to anti-PD-1 therapy (X) between low- and high-TLS clusters in Metastatic Urothelial carcinoma cohort (GSE176307). (Y) The box diagram showed the differences of drug sensitivity (IC50) to targeted therapy between high-and low-TLS score clusters. $*p < 0.05$. $**p < 0.01$. $***p < 0.001$. $****p < 0.0001$.

### Association of TLS score with immune infiltration

Spearman correlation analysis established a positive correlation between the TLS score and the infiltration of 23 different immune cells. This correlation suggested an augmented immune component presence in the TIME of patients within the high-score cluster, potentially indicating a more favorable immune prognosis (Fig. 4K, $p < 0.05$). Building upon these findings, we delved deeper into the association of TLS with cytokine-chemokine interaction networks. Notably, the high-score TLS cluster demonstrated significant enrichment in chemokines and their receptors, interleukins, and interferons along with their respective receptors (Fig. 4L). The hallmark pathway analysis revealed a significant positive correlation 12 of the TLS score with pathways indicative of immune activation. These pathways included interferon gamma response, interferon alpha response, inflammatory response, and IL-2 STAT5 signaling. Conversely, an inverse correlation was observed between the TLS score and pathways typically associated with malignancy, such as peroxisome proliferation, MYC target activation, oxidative phosphorylation, fatty acid metabolism, and DNA repair mechanisms (Fig. 4M). These findings underscored the strong association of the TLS score with predominantly immune-activated pathways.

### Association of TLS score with MSI, TMB, CSC, and somatic mutations

Cancer stem cells (CSCs), which were acknowledged for their significant roles due to their selfrenewal capacity and differentiation potential (*Nassar & Blanpain, 2016*). We delved into the relationship between the TLS score and CSC indices, including mDNAsi and mRNAsi. Our analysis, as depicted in Fig. 5A, revealed a notable negative linear correlation between the TLS score and the mDNAsi index (R = −0.18, $p = 6e−04$). This trend was further demonstrated by the lower mDNAsi observed in the high TLS score cluster compared to the low TLS score cluster ($p < 0.01$; Fig. 5B). Similarly, Fig. 5D presented a negative linear correlation between the TLS score and the mRNAsi index (R = −0.63, $p = 0$), with the high TLS score cluster exhibiting significantly lower mRNAsi than its low-score counterpart ($p < 0.0001$; Fig. 5E). Heat maps further corroborate these findings, illustrating the inverse relationship between both mDNAsi and mRNAsi index values and the TLS score (Figs. 5C, 5F). These insights suggested that COADREAD patients with a higher TLS score may exhibit reduced stem cell characteristics and diminished cell differentiation capabilities.

In the context of microsatellite instability (MSI), a promising genomic biomarker for gauging 13 patient responsiveness to immunotherapy (*Vilar & Gruber, 2010*), our findings revealed a significantly higher MSI in the high TLS score cluster compared to the low ($p < 0.05$; Fig. 5H). Spearman correlation analysis further substantiates this, indicating a positive association between MSI and TLS score (R = 0.18, $p$ = 7e−04; Fig. 5G). The positive correlation between MSI expression and TLS score was also visually represented in a heat map (Fig. 5I). This pattern suggested a potential link with enhanced immunotherapy efficacy.

Regarding tumor mutational burden (TMB), which served as an indicator of cancer mutation volume and has clinical relevance with ICIs outcomes (*Chan et al., 2019*), our data analysis from TCGACOADREAD cohorts indicated a higher TMB in the high TLS score cluster than in the low score cluster ($p < 0.001$; Fig. 5K). Spearman's analysis corroborated a positive relationship between TMB and TLS score (R = 0.2, $p$ = 1e−04; Fig. 5J), with a heat map further illustrating this correlation (Fig. 5L). These findings were particularly noteworthy for patients with higher TMB who typically show more benefit from ICIs.

Our analysis extended to the somatic mutation landscape within the two distinct TLS score clusters. In the high TLS score cluster, the predominant mutated genes were *TP53*, *TTN*, *APC*, *KRAS*, *SYNE1*, *MUC16*, *PIK3CA*, *OBSCN*, *FAT4*, and *RYR2*. Conversely, the low TLS score cluster was characterized by mutations primarily in *APC*, *TP53*, *KRAS*, *TTN*, *SYNE1*, *PIK3CA*, *FAT4*, *MUC16*, *OBSCN*, and *MUC4*. Notably, the mutation frequency across these genes was more pronounced in the high TLS score cluster, as illustrated in Figs. 5M and 5N. Additionally, a forest plot detailed the top 12 genes exhibiting the most significant differences in mutation frequency between the high and low score clusters (Figs. 5O). This led to the inference that gene mutation frequency in the high TLS score cluster was not only higher but also more14 diverse. Detailed SNV, CNV, and methylation profiles of prognostic TLS-DEGs in COADREAD patients were depicted in Figs. S2 and S3. Analysis revealed significant disparities in recurrence/metastasis, stage, and fustat between the low- and high-TLS score clusters (Figs. 5P–5S).

### Interplay of TLS score and immune checkpoints expression

Subsequent research focused on the expression levels of immune checkpoints in relation to TLS scores. Elevated expression of key immune checkpoints, including *PDCD1* (*PD-1*), *CD274* (*PD-L1*), *CTLA4*, *TIGIT*, and *LAG3*, was predominantly observed in the high-score cluster, as depicted in Fig. 5T. This trend suggested that TLS score was potentially served as a predictive marker for immunotherapy efficacy.

### Evaluating TLS score as a predictor for PD-L1 blockade immunotherapy

In the realm of T-cell immunotherapy, a promising approach for cancer treatment, we explored the significance of the TLS score in COADREAD, building on prior findings and our current results (*Waldman, Fritz & Lenardo, 2020*). Within the GSE135222 cohort, a notable observation was the higher progression rate (100%) in patients with lower TLS scores compared to those with higher scores (75%) (Fig. 5V). Furthermore, a markedly

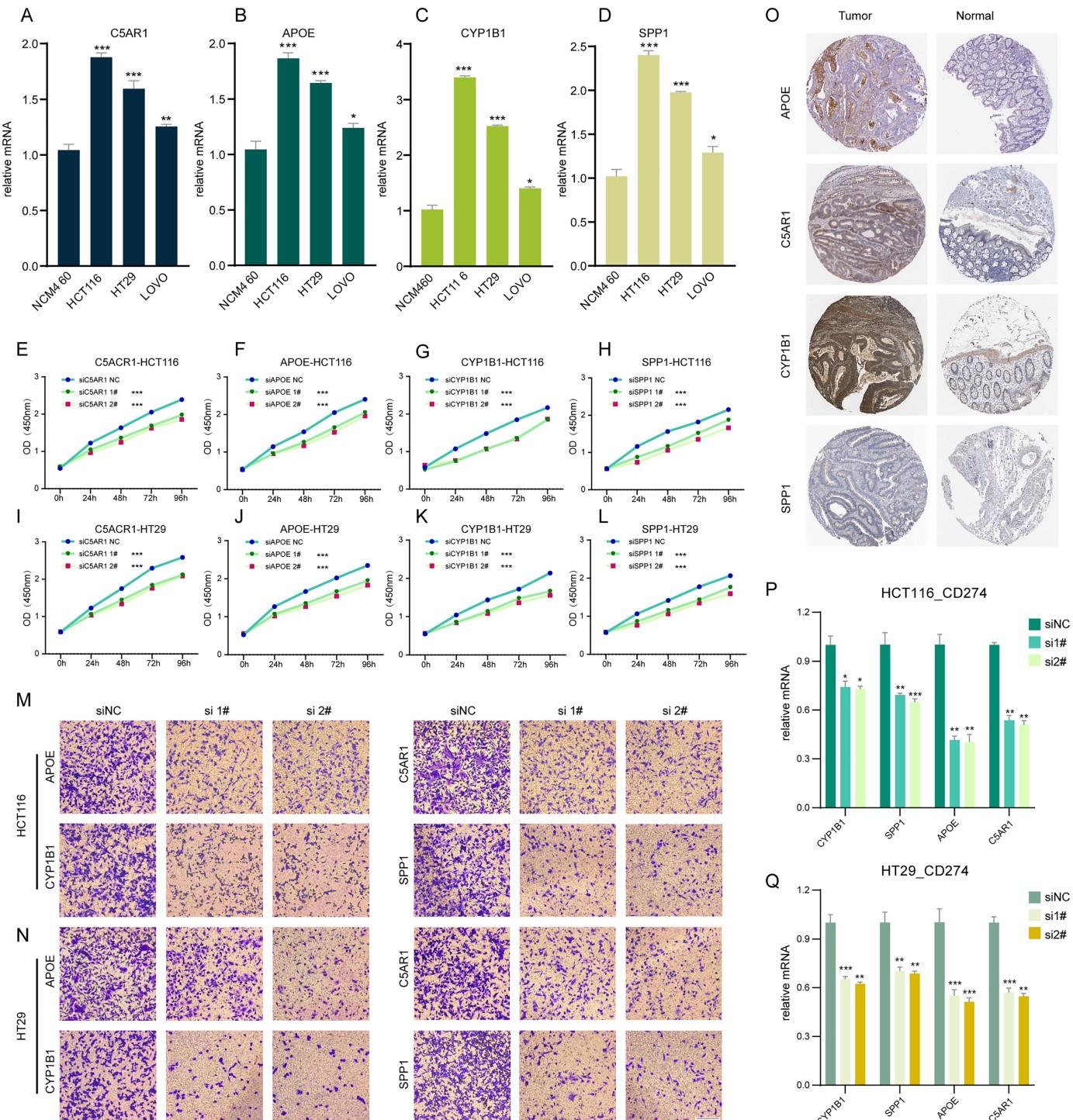

**Figure 6 Cell experiment and tissue expression verification.** Validation of the expression of C5AR1 (A), APOE (B), CYP1B1 (C), SPP1 (D) in a normal colon cell line (NCM460) and three COADREAD cell lines (HCT116, HCT29, and VOLO) by qRT-PCR. (EL) CCK8 assay. After knockdown of C5AR1 (E, I), APOE (F, J), CYP1B1 (G, K), and SPP1 (H, L), the HCT 116 cells and HT 29 cells showed significant reduction in viability separately. The invasion capacity of HCT116 cells and HCT29 cells decreased significantly after APOE, C5AR1, CYP1B1, and SPP1 knockdown (M-N). Immunohistochemistry showing the protein expressions of APOE, C5AR, CYP1B1, and SPP1 based on the Human Protein Atlas (HPA) database (O). After knockdown of APOE, C5AR, CYP1B1, and SPP1, the HCT 116 cells and HT 29 cells showed significant reduction of PD-L1 expression separately (P and Q).

prolonged progression-free survival (PFS) was evident in patients with higher TLS scores (Fig. 5U, $p$ = 0.0061). Analyzing the GSE176307 cohort, which comprised 90 patients treated with anti-PD-L1 receptor blockers, revealed a spectrum of responses ranging from complete response (CR) and partial response (PR) to stable disease (SD) and progressive disease (PD). Significantly, patients exhibiting CR/PR had elevated TLS scores compared to those with SD/PD (Fig. 5X). In this cohort, patients in the high-TLS cluster demonstrated substantial clinical benefits and notably longer overall survival (OS) than their low-TLS counterparts (Fig. 5W, $p$ = 0.018).

### Analysis of drug susceptibility in relation to TLS score

Subsequent to establishing the TLS score as an effective predictor for immunotherapy response, our study delved into the variances in IC50 values for various targeted anticancer drugs across low- and high-TLS clusters. Intriguingly, patients within the high-TLS cluster exhibited notably lower IC50 values for a range of anticancer agents, namely A.770041, AP.24534, AG.014699, AICAR, ABT.263, and AMG.706 (Fig. 5Y). These findings underscored the potential utility of the TLS score as a guiding metric in selecting appropriate anticancer drugs.

### Impact of TLSRGs on colorectal cancer cell behavior in vitro

In our investigation, RT-q PCR was utilized to ascertain the expression levels of *C5AR1*, *APOE*, *CYP1B1*, and *SPP1* in a standard colon cell line and three colorectal cancer cell lines (Figs. 6A–6D). These four genes were markedly upregulated in cancer cells compared to normal cells. We employed siRNA techniques targeting *C5AR1*, *APOE*, *CYP1B1*, and *SPP1*. This was conducted in HCT116 and HT29 cell lines, selecting siRNA-1 and siRNA-2 for further analysis based on their transfection efficiency exceeding 70% (Figs. S4A–S4H). A CCK8 assay was subsequently performed to evaluate cell proliferation. Results demonstrated that the knockdown of *APOE*, *C5AR1*, *CYP1B1*, and *SPP1* significantly impeded the proliferation of both HCT116 and HT29 cells (Figs. 6E–6L). The Transwell assay was conducted to assess cellular invasion capabilities. Notably, the suppression of *APOE*, *C5AR1*, *CYP1B1*, and *SPP1* led to a marked reduction in the invasion capacity of HCT116 and HT29 cells (Figs. 6M and 6N). Furthermore, an unexpected observation was the considerable decrease in *PD-L1* expression in both HCT116 and HT29 cells following the knockdown of these genes. This reduction in *PD-L1* expression could hint at the potential of combining TLS inducers with PD-L1 inhibitors as a therapeutic strategy (Figs. 6P and 6Q). Additionally, the immunohistochemical data from the HPA database revealed an elevated protein expression of *APOE*, *C5AR1*, *CYP1B1*, and *SPP1* in the stroma of COADREAD tissue (Fig. 6O).

## DISCUSSION

Globally, the prevalence of COADREAD, a leading form of cancer, is escalating in numerous nations. Recent endeavors in COADREAD management have been substantial; however, the disease's heterogeneous nature and its aggressive tendencies continue to pose challenges in prognostic evaluations (*Morgan et al., 2023*; *Morris et al., 2023*;

*Ludford et al., 2023*). Identifying new biomarkers is thus imperative and urgent, offering a pathway to tailor patient-specific treatments and enhance prognostic accuracy.

While immune checkpoint inhibitors and chimeric antigen receptor T cell therapies are established as safe for various cancer types, their efficacy limitations underscore the need to explore additional cellular pathways for more effective treatment strategies (*Marisa et al., 2018*). Recent studies have linked the development of tertiary lymphoid structures (TLS) with improved clinical outcomes across several cancer types, and these structures are believed to enhance the efficacy of immunotherapies (*Groeneveld et al., 2021*; *Ruffin et al., 2021*; *Vanhersecke et al., 2021*). In colorectal cancers, which exhibit significant tumor heterogeneity, only certain molecular subtypes respond to immunotherapy (*André et al., 2020*; *Ludford et al., 2023*; *Gurjao et al., 2019*). Therefore, TLS could potentially provide insights into predicting and selecting COADREAD patients for immunotherapy, and in addressing immunotherapy resistance, particularly in groups with traditionally low response rates.

In contrast to bulk RNA-sequencing, which gauges average gene expression levels in cell populations, scRNA-seq has risen as a pivotal technique. It facilitates the delineation of cellular subpopulations, pinpointing distinctive biomarkers, and understanding the heterogeneity across different cell types in an array of cancers (*Kotsiliti, 2022*).

Thus, this study undertook an extensive analysis combining both bulk RNA-seq and scRNA-seq. Our approach encompassed clustering analysis, examination of TLS-associated DEGs, assessment of immune infiltration and mutation landscapes, and screening for prognostic genes. This comprehensive methodology enabled us to formulate a TLS score, demonstrating substantial predictive accuracy for immunotherapy effectiveness in COADREAD. Initially, our scRNA-seq analysis on COADREAD samples delineated two distinct TLS score clusters. In the cluster with elevated scores, we noted a significant enrichment of critical immune activation markers, such as inflammatory response, interferon-gamma and alpha responses, and TNFA signaling through NFKB pathways, aligning with earlier findings (*Yenyuwadee et al., 2022*). Further, a higher prevalence of monocytes, macrophages, and CD8+ T cells was evident in the high score group compared to the low-score group, which predominantly consisted of malignant cells and fibroblasts. This distribution implied that the high-score group harbors a more robust anti-tumor immune cell congregation, thereby intensifying anti-tumor responses. This observation was in concordance with recent literature suggesting TLS's role in invigorating antitumor immunity and boosting immunotherapy efficacy (*Cabrita et al., 2020*; *Groeneveld et al., 2021*; *Chelvanambi et al., 2021*), reinforcing the concept of TLSs as indicative of "hot tumors".

Subsequently, our investigation led to the identification of 12 TLSRGs with prognostic significance. Utilizing these genes, we stratified COADREAD patients into two distinct clusters. Our study distinguished 189 DEGs between the two identified clusters, leading to the isolation of 16 genes intimately linked with overall survival (OS). we segregated COADREAD patients into two gene clusters based on these 16 prognostic genes. Leveraging this groundwork, we crafted a TLS score model using GSVA. This model incorporated 16 pivotal TLS-DEGs: *HOXC6*, *SFRP2*, *CYP1B1*, *GAS1*, *SPOCK1*, *SPP1*,

*POSTN, MARCO, VSIG4, THBS2, OLR1, COL10A1, FAP, GZMB, C5AR1*, and *APOE.* Their prognostic relevance in various malignancies was further corroborated by previous research.

The role of *HOXC6* in nonmetastatic CRC is noteworthy, its heightened expression correlates significantly with increased immunogenicity (*Qi et al., 2021*). Within an aging microenvironment, *sFRP2* has been identified as a driver of melanoma metastasis and resistance to therapies (*Kaur et al., 2016*). The enzyme *CYP1B1* played a crucial role in the advancement and progression of castration-resistant prostate cancer (*Lin et al., 2022*). Interestingly, *GAS1* impeded colorectal tumorigenesis through WNT signaling, influencing CD143+ cancer-associated fibroblasts (*Lin et al., 2022*). As a potential cancer prognostic marker, *SPOCK1* enhanced proliferation and metastasis in gallbladder cancer cells *via* the PI3K/AKT pathway (*Shu et al., 2015*). The protein *SPP1*, known for its upregulation of *PD-L1*, contributed to lung cancer's ability to evade immune detection (*Zhang et al., 2017*). *POSTN*, by augmenting M2 macrophages and cancer-associated fibroblasts, facilitated ovarian cancer metastasis (*Lin et al., 2022*). In lung cancer, targeting *MARCO* on immunosuppressive macrophages has shown promise, hindered regulatory T cells while bolstering cytotoxic lymphocyte functions (*La Fleur et al., 2021*). Finally, *VSIG4*, which regulated immune homeostasis by modulating complement pathways and T-cell differentiation, presented a double-edged sword. While it curbed immune-mediated inflammatory diseases, it simultaneously aided in cancer progression, positioning it as a novel target in cancer immunotherapy (*Liu et al., 2023*). In the context of early-stage lung adenocarcinoma, *THBS2*+ cancer-associated fibroblasts have been pinpointed as crucial facilitators of the disease's aggressiveness through multi-scale integrative analyses (*Yang et al., 2022*). Research revealed that silencing *OLR1* attenuates glycolytic metabolism, thereby inhibiting proliferation and chemoresistance in colon cancer cells (*Zhao et al., 2021*). *COL10A1* has been identified as a promoter of gastric cancer invasion and metastasis, primarily through the process of epithelial-to-mesenchymal transition (*Li et al., 2018*). *FAP* appeared to exacerbate malignancy outcomes through various mechanisms, including extracellular matrix remodeling, intracellular signaling, angiogenesis, epithelial-to-mesenchymal transition, and immunosuppression (*Fitzgerald & Weiner, 2020*). *GZMB* stood out as a primary indicator of cytotoxic T-cell activity, with *GZMB*+ T-cells being pivotal in antitumor immunity (*Bassez et al., 2021*). The intracellular *C5a/C5aR1* complex was found to stabilize β-catenin, thereby facilitating colorectal tumorigenesis (*Ding et al., 2022*). Furthermore, *APOE*, secreted by prostate tumor cells, has been linked to cellular senescence and is associated with adverse prognoses (*Bancaro et al., 2023*). This study presented, for the first time, an extensive exploration of the roles of these TLS signature genes in COADREAD.

Subsequent findings indicated that individuals with higher TLS scores potentially derive greater benefit from anti-PD-1 therapies and targeted treatment approaches. Intriguingly, we observed a contrasting prognosis associated with TLS scores in patients undergoing immunotherapy *vs.* those who were not. This aligns with the observed higher rates of immune cell infiltration, immune checkpoint expression, MSI index, TMB index, and somatic mutations in the high-TLS score cohort. Furthermore, a significant positive

correlation was noted between most immune activation-related pathways and TLS scores. Existing literature supported the notion that both immunotherapy and chemotherapy can promote the development of tertiary lymphoid structures, including the proliferation of CD8+ T cells, thereby bolstering anti-tumor immunity (*Cabrita et al., 2020*; *Helmink et al., 2020*; *Vanhersecke et al., 2021*; *Lee et al., 2022*). This aligned with our RNA-seq data analysis. Similarly, our single-cell data model corroborated this, indicating enhanced anti-tumor immune cell infiltration and pathway activation in the high-score group. These insights underscored the pivotal role of TLS status in determining the efficacy of immunotherapy and other immune-based treatments.

This study provides important insights into the prognostic and therapeutic potential of TLS-related gene profiles in colorectal cancer. However, several limitations must be acknowledged to fully contextualize the findings and guide future research efforts. A primary limitation is the reliance on RNA sequencing data without direct histopathological validation of TLS presence. While TLS-related gene signatures were identified, their correlation with actual TLS structures in the tumor microenvironment was not confirmed. Future studies should incorporate histopathological validation, such as immunohistochemistry or immunofluorescence, to directly assess TLS formation and establish a clearer link between gene signatures and TLS presence.

Additionally, while this study provides a comprehensive analysis of TLS-related gene signatures, it does not explore the underlying mechanisms that drive TLS formation or their functional impact on immune responses. Mechanistic insights into how TLSs form and interact with immune cells within the tumor microenvironment are crucial for understanding their role in modulating immune activity and influencing patient outcomes. Future research should focus on functional assays and mechanistic studies using *in vitro* and *in vivo* models to investigate these pathways.

Another limitation stems from the reliance on publicly available datasets from GEO and TCGA, which introduces potential biases. These datasets may not fully represent the diversity of colorectal cancer populations, particularly in terms of ethnicity, disease stage, and treatment history, which could limit the generalizability of the findings. Validation in more diverse patient cohorts, ideally through prospective multicenter studies, will be necessary to ensure the applicability of the TLS score across broader clinical settings.

Moreover, the study did not explicitly control for confounding variables such as tumor stage, treatment history, or patient demographics, which could influence the TLS score's correlation with immunotherapy efficacy. These factors might have affected the outcomes and should be considered in future research. Stratifying patients or adjusting for confounders in statistical analyses will help ensure more accurate conclusions about the clinical utility of the TLS score.

A further limitation is the absence of functional validation, which makes some conclusions speculative. Although the bioinformatics analysis suggests that TLS presence correlates with immune responses, direct experimental validation is needed to confirm these findings. Future studies should employ functional assays, such as co-culture systems

or animal models, to test the biological role of TLSs in modulating immune responses and treatment outcomes.

The clinical relevance of the TLS score remains somewhat unclear, as we did not fully clarify its association with key biomarkers such as immune responses, tumor mutational burden (TMB), microsatellite instability (MSI), and specific gene signatures. While immune response markers were shown to correlate with higher TLS scores, the relationships with TMB and MSI need further exploration. Future research should refine the TLS score, better delineating its associations with these biomarkers and validating its use in clinical decision-making.

The complexity of the bioinformatics approach used in this study may also pose challenges for replication. However, we have provided all necessary methodological details, along with publicly available data and code, to ensure that other researchers can replicate and build upon these findings.

Additionally, the findings of this study have not been validated in clinical trials, limiting their immediate applicability in clinical practice. Prospective clinical trials will be essential to evaluate the TLS score's ability to predict treatment responses and guide personalized therapy in colorectal cancer patients. Such trials would also help assess the long-term utility of TLS-related biomarkers in predicting recurrence and survival outcomes, which were not covered by the current study.

Finally, while this study offers a broad assessment of TLS-related gene profiles, the global nature of the analysis may dilute the focus. A more targeted approach, investigating specific hypotheses related to TLS function or its relevance in immunotherapy, would yield more actionable conclusions. Future research should aim for a deeper exploration of particular aspects of TLS biology that have the most clinical significance.

In conclusion, while this study provides valuable insights into TLS-related gene profiles in colorectal cancer, addressing these limitations in future research will significantly enhance the robustness and clinical relevance of TLS as a prognostic and therapeutic marker.

## CONCLUSIONS

In this study, we elucidated the distinct features of TLS within the COADREAD framework. Utilizing a quadratic clustering-based TLS score model, our approach enabled the assessment of TLS dynamics at an individual level. This method also facilitated predictions regarding the effectiveness of immunotherapy and other immune-based treatments in COADREAD cases. Consequently, a deeper comprehension of TLS patterns, combined with the application of the TLS scoring system, held promise for guiding clinical decisions and enhancing the prognosis for patients with colorectal cancer.

## ACKNOWLEDGEMENTS

We are grateful to Lianxiang Luo for technical assistance and advice.

### Funding

The study was supported by the Medical Innovation Special Project of Shanghai Municipal Science and Technology Commission (22Y31920103). The funders had no role in study design, data collection and analysis, decision to publish, or preparation of the manuscript.

### Grant Disclosures

The following grant information was disclosed by the authors:
Medical Innovation Special Project of Shanghai Municipal Science and Technology Commission: 22Y31920103.

### Competing Interests

The authors declare that they have no competing interests.

### Author Contributions

- Jinglu Yu conceived and designed the experiments, performed the experiments, analyzed the data, prepared figures and/or tables, authored or reviewed drafts of the article, and approved the final draft.
- Yabin Gong analyzed the data, authored or reviewed drafts of the article, and approved the final draft.
- Xiaowei Huang performed the experiments, authored or reviewed drafts of the article, and approved the final draft.
- Yufang Bao conceived and designed the experiments, prepared figures and/or tables, authored or reviewed drafts of the article, and approved the final draft.

### Data Availability

The raw measurements are available in the Supplemental Files.

### Supplemental Information

Supplemental information for this article can be found online at http://dx.doi.org/10.7717/peerj.18401#supplemental-information.

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
