# Peer review of "Prognostic and therapeutic potential of gene profiles related to tertiary lymphoid structures in colorectal cancer"

_PeerJ, doi:10.7717/peerj.18401_

## Round 0.1 · original submission · Minor Revisions

The experts found your study interesting and provided suggestions for improving the clarity of the manuscript that should be implemented. In addition, because there is no validation in the clinical settings, it is difficult to evaluate the applicability of the findings for therapy. They propose that you change the title to make clear that only gene expression profiles have been analyzed. Reviewer 3 requested that you develop on which of TLS score: based on immune responses, TMB status, MSI or other specific gene signatures is relevant based on your analysis. Finally, add all the limitations as highlighted by the reviewers.

Reviewer 1 ·

Basic reporting

In Figure 1, due to the vast number of analyses and several used databases, I would suggest, if possible, adding the used databases per analysis either in the text within the figure or in the figure legend to help the reader to structure the used methods and used databases. If added in figure legend, partitioning the boxes into A, B, C etc. might also ease the readability.

Part of the figures, especially Figure 5, has so much data, that it might be impossible to read from the paper version. Are all the figures necessary in Figure 5? Could the Figure 5 be separated into two figures or could part of the analyses be located in the supplementary section?

Lines 72-74 are missing a reference (I assume it is ref no 29?). Please add references, since this is a major point in the article.

In some figures, for example in Figure 3 F-I, the coloring of projects GSE17538 and GSE39582 are so similar that they are hardly distinguishable. Please correct.

In figure legend 3, the section D is before C. Also, the text in section F has a missing capital letter. Please correct.

There are spelling mistakes in the hyperlinks in lines 94, 95, 147 and 156. I assume it should be https://...? Please check all the hyperlinks.

Lines 496-497: the references are presented in the wrong font without a superscript.

Line 438: there is a period after “recurrence” instead of a comma, I assume. Please correct. The same mistake is in the figure legend of Figure 5, after the word “recurrence”.

Experimental design

No comment.

Validity of the findings

As this is a study based on RNA sequencing data, and no analysis of TLSs were performed from the histopathological slides, I would suggest this should be reflected in the title of the article. For example: “Prognostic and therapeutic potential of gene profiles RELATED to tertiary lymphoid structures in colorectal cancer”, might better express the study design. Also, I think the lack of actual TLS analysis from the used samples should be mentioned in the limitations.

Reviewer 2 ·

Basic reporting

This study presents a comprehensive analysis of Tertiary Lymphoid Structures (TLS) in colorectal cancer, highlighting their potential as prognostic biomarkers and therapeutic targets. By employing single-cell sequencing and a novel TLS quantification system, the authors successfully established a TLS score that correlates with immunotherapy efficacy in a large cohort of patients. However, the study has several weaknesses: it lacks a thorough exploration of the mechanisms underlying TLS formation and their direct impact on immune responses, which limits the understanding of how these structures function within the tumor microenvironment. Additionally, while the identification of distinct molecular subtypes based on Tertiary Lymphoid Structure-related Genes (TLSRGs) is valuable, the practical implications for clinical applications and treatment strategies remain unclear. The findings would benefit from validation in diverse patient populations and consideration of potential confounding factors that could influence TLS presence and function. Overall, while the research contributes valuable insights into personalized medicine in colorectal cancer, addressing these weaknesses would enhance its impact and applicability.

Experimental design

The experimental design of the study, while robust in utilizing single-cell sequencing and bulk RNA sequencing, has some notable weaknesses. First, the reliance on datasets from GEO and TCGA may introduce biases, as these datasets may not fully represent the diversity of colorectal cancer patient populations, potentially limiting the generalizability of the findings. Additionally, the study does not clearly outline how confounding variables, such as tumor stage, treatment history, or patient demographics, were controlled or accounted for in the analysis. This oversight could affect the validity of the conclusions drawn regarding TLS scores and their correlation with immunotherapy efficacy. Furthermore, the lack of functional assays to directly assess the biological implications of TLS and their interaction with immune cells is a significant gap; without such data, the causal relationships between TLS presence and treatment outcomes remain speculative. Overall, enhancing the experimental design by incorporating more diverse patient samples, controlling for confounding factors, and including functional studies would strengthen the study’s conclusions.

Validity of the findings

The findings of this study appear valid, as they are supported by a robust analytical framework that incorporates single-cell RNA sequencing and bulk RNA-seq data from large patient cohorts. The identification of distinct molecular subtypes based on Tertiary Lymphoid Structure-related Genes (TLSRGs) and the establishment of a TLS score, which correlates with immunotherapy outcomes, strengthens the reliability of the results. Additionally, the use of multiple independent cohorts for validation enhances the credibility of the conclusions drawn regarding TLS as potential prognostic biomarkers and therapeutic targets in colorectal cancer. However, further validation in clinical settings and exploration of the underlying mechanisms of TLS interactions in the tumor microenvironment are necessary to fully establish their clinical relevance.

Additional comments

The study's findings are promising but could benefit from further validation in clinical trials to assess their applicability. Additionally, exploring the molecular mechanisms of TLS, ensuring diverse patient representation, conducting long-term follow-up studies, and integrating TLS scores with other biomarkers could enhance the overall understanding and utility of TLS in colorectal cancer treatment.

Reviewer 3 ·

Basic reporting

The Language of the article is clear, but inclusion of multiple assessments makes it confusing to understand.

Experimental design

Research question is not well defined and focused, rather offers global assessment of TLS in colorectal cancer.
Authors have used extensive bioinformatics tools to assess publicly available RNA sequencing data. Methods are well described but will not be easy to replicate.

Validity of the findings

Authors globally assessed previously available RNA sequence data with a focus on TLS but without a meaningful study plan that should lead to experimental validations. Thus, conclusions are speculative. Experimental data is required to support speculative conclusions.

Additional comments

The study “Prognostic and therapeutic potential of gene profiles from tertiary lymphoid structures in colorectal cancer by Jinglu Yu et.al. used retrospectively publicly available sequencing data for comprehensive bioinformatics assessment to examine TLS. While this descriptive computational study is technically sound but fails to provide any conclusive clinically applicable decision-making.
Every element of result section is non conclusive as they only either show impact or evaluation or interplay among TLS and many factors (such as MSI, mutations, Immune cells, gene subtypes, hallmark pathways, drug susceptibility etc.) they analyzed.
Authors described about developing TLS score at many instances based on their multiple assessment but failed to clarify their significance in discussion. It not clear TLS score based on Immune responses, TMB status, MSI or which specific gene signatures is relevant ?

---

## Round 0.2 · Minor Revisions

Thank you for addressing most of Reviewers' concerns.

One point still needs to be clarified.

In the rebuttal letter addressing one of the concerns of Reviewer 3 it is stated: “To address this, we have provided more detailed descriptions of the software, parameters, and datasets used in the study and ensured that all code and data are publicly available to facilitate replication.”

However:

1. There are no changes in the methods section and the data used are publicly available
2. There were 28 codes with the submission and none have been newly added. If the 28 codes are sufficient to replicate the analysis it is not clear why in the discussion, Line 382 of tracked changes version, it is stated: “To ensure that other researchers can replicate and build upon these findings, future studies should provide more detailed methodological documentation and make data and code publicly available through open repositories.”

Therefore please state whether the 28 codes provided are sufficient to replicate the analysis or add the codes that are missing and modify the sentence in the discussion.

---

## Round 0.3 · accepted · Accept

Thank you for addressing last concern. I had forgot to mention that a particle was missing in the title that now reads: "Prognostic and therapeutic potential of gene profiles related to tertiary lymphoid structures in colorectal cancer".